# How Important Are Fog and the Cloud Forest as a Water Supply in Eastern Mexico?

Antonio Sánchez-Falfan [1,2], Manuel Esperón-Rodríguez [3], Juan Cervantes-Pérez [2], Monica Ballinas [1,4] and Victor L. Barradas [1,*]

1   Laboratorio de Interacción Planta Atmosfera, Instituto de Ecología, Universidad Nacional Autónoma de México, Mexico City 04510, Mexico
2   Facultad de Instrumentación y Ciencias Atmosféricas, Universidad Veracruzana, Xalapa 91090, Mexico
3   Hawkesbury Institute for the Environment, Western Sydney University, Locked Bag 1797, Penrith, NSW 2751, Australia
4   Centro de Investigación en Ciencias de Información Geoespacial, A.C., Mexico City 14240, Mexico
*   Correspondence: vlbarradas@ecologia.unam.mx

**Abstract:** The water balance is the volume of water flowing through the hydrological cycle, and one of its main components is fog. Fog is considered a type of low-lying cloud and is heavily influenced by water bodies, topography, and wind conditions. Fog incorporates water from the atmosphere to the terrestrial surface and for some ecosystems (e.g., cloud forests) represents a great water contribution. In this work, we aimed to answer the following questions: (1) What is the fog-water contribution to the water balance? (2) How does the presence of vegetation affect the water supply to the ecosystem? We took as a case study the Central Mountain Region of Veracruz, in eastern Mexico, and measured components of the water balance; this included precipitation (gross and net), fog water, interception, transpiration, evapotranspiration, and condensation, and we estimated water gain and loss of the water balance. We registered 510 precipitation events distributed throughout the year with three peaks (October, May, and January). Of these, 386 were fog events, 41 were rain events, and 83 were events combining fog and rain. Fog had a substantial contribution of water to the system, with a volume 22 times greater than that of rain (4311.14 mm vs. 197.5 mm). From the total water gain, the highest proportion (91%) was contributed by fog interception. Fog was considered a constant source of moisture throughout the year; however, water intercepted during the dry season was higher (56%) compared with the wet season (36%). Our results highlight the importance of the fog as a source of water for the region.

**Keywords:** forest canopy; interception; precipitation; vegetation cover; water balance

## 1. Introduction

The water balance is defined as the volume of water flowing through the hydrological cycle, and usually, it is composed of components that are considered storage units [1]. Changes in these components can be quantified to assess how the balance of the hydrological cycle is affected [2]. Water may evaporate or be intercepted by plants when it is precipitated. Precipitation can be classified into two types, gross precipitation ($P_G$) and net precipitation ($P_N$) (sensu [3–6]). Net precipitation ($P_N$) is the amount of water that reaches the ground by direct or indirect (via vegetation) dripping, or runoff. Depending on the vegetation type and canopy, $P_N$ varies broadly. For example, in a forest with a high leaf area index, a considerable part of the precipitation does not reach the ground because of interception (i.e., the part of the rainfall that is intercepted by vegetation) [7,8] or evaporation (i.e., the process of surface water absorbing heat energy from the sun or from warm air and changing from liquid to gas) [3,6]. As for $P_G$, this is the precipitation that reaches the top of the vegetation; however, the vegetation surface may have no effect or can decrease or increase the occurrence, quantity, or duration of a precipitation event [9,10]. Although

forests generally do not increase $P_G$ (e.g., [11,12]), there are some relevant exceptions in the humid tropics. For instance, in extensive forested areas of the Amazon basin, Salati et al. [13] showed the existence of humidity recycling through evapotranspiration, condensation, and precipitation processes on a large scale. Furthermore, in some mountainous areas often covered by clouds and fog, the presence of cloud forests can increase $P_G$ through fog catchment mechanisms [14–16].

Fog incorporates water from the atmosphere to the terrestrial surface [17], and in some regions, it can occur throughout the year due to the orographic rise of air and the later adiabatic cooling [18]. One of the main effects of fog is to wet plant foliage, which reduces the transpiration rate and water evaporation from the soil due to conditions of high humidity and low air temperature [19,20]. In addition, fog represents a significant source of additional moisture in ecosystems with high fog incidence, such as those located in the cloud belt of mountain slopes exposed to prevailing winds [19,21,22].

In those ecosystems, vegetation (trees in particular) incorporates to the system significant amounts of water. However, these amounts are usually omitted and neglected in water balance analyses [23] because of the low proportion of water that reaches the soil; that is, precipitation can decrease in areas with vegetation cover because of the storage capacity of the canopy [19]. Furthermore, this water is not measured by standard-type rain gauges because it depends strongly on the dominant vegetation and canopy characteristics [21]. Wet leaves, for example, store water that does not reach the soil because it evaporates. This storage depends on leaf traits (e.g., texture and size), as well as canopy density [24]. Nevertheless, the water intercepted by trees and the water that precipitates can meet the water demand of vegetation [25].

In this work, we aim to highlight the importance of the relationship between fog and vegetation, as both components play key roles in the ecosystemic water balance. To assess this relationship, we took as a case study the Central Mountain Region of Veracruz (CMRV), Mexico. The CMRV is a region with high fog frequency [19,23] and is characterized by the presence of the cloud forest, the most threatened terrestrial ecosystem in Mexico [26] (CONABIO 2010). In this region, climate change and deforestation have led to a decrease in precipitation [27–29] representing a loss in the water balance and in the water supply to surrounding urban areas [19]. Despite these changes in the region, little is known about the importance of fog in the water balance and its relationship with the local vegetation. Our objective was to answer two questions: (1) What is the water contribution of the fog to the water balance? (2) How does the presence of vegetation affect this water supply?

## 2. Materials and Methods

### 2.1. Study Area and Selected Species

Our study area was the Central Mountain Region of Veracruz (CMRV) in Mexico, specifically La Joya, located in the municipality Acajete (19° 38′ N, 97° 05′ W, 2421 m above sea level [asl]; Figure 1). The CMRV is part of the Neovolcanic Ridge and the Sierra Madre Oriental. Abrupt topography is the main characteristic of this region with a pronounced altitudinal gradient from the sea level up to 5500 m asl, in a distance of 100 km. As a result, contrasting vegetation types can be found, from tropical montane cloud forests to semi-arid and arid shrub communities [23,30]. The average annual temperatures range between 10 and 29 °C, and the annual precipitation ranges from 600 to 3000 mm [27]. Climate classification subtype is Aw (Tropical Savanna Climate).

All measurements were performed from August 2006 to August 2007. Five tree species were selected according to their dominance within the CMRV, and the number of individual trees selected varied across species: *Alnus acuminata* Kunth (four individuals), *Cupressus benthamii* Endlicher (1847) (one individual), *Crataegus mexicana* Moc. & Sessé ex DC. (one individual), *Pinus ayacahuite* Ehren (two individuals), and *Pinus patula* Schiede ex Schltdl. & Cham (five individuals) (Supplementary Table S1).

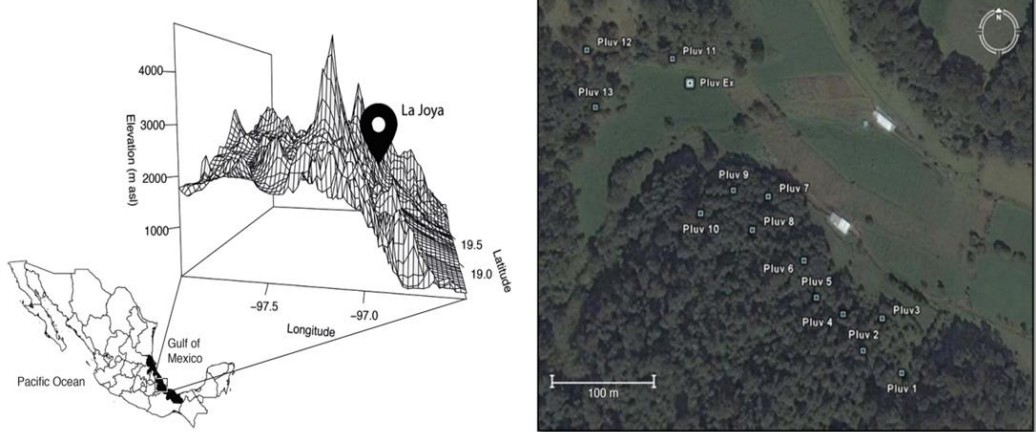

**Figure 1.** Location of La Joya in the Central Mountainous Region of Veracruz, Mexico, and location of the rain gauges in the study area. The external rain gauge is named "Pluv Ex".

### 2.2. Interception of Water by the Canopy

Interception is defined as the amount of precipitated water retained by the vegetation and then evaporated [3] and is given by the equation [31]

$$I_T = P_G - (P_T + P_F) = P_G - P_N \tag{1}$$

where $I_T$ is the total interception, $P_G$ is the gross precipitation, $P_N$ is the net precipitation (produced by $P_G$ or fog retained by the canopy and subsequently precipitated), $P_T$ is the throughfall precipitation, which is the precipitation that passes through the foliage, and $P_F$ is the stemflow precipitation, which is the water that flows through the trunk into the soil.

The gross precipitation ($P_G$) was measured with a rain gauge located in an open area with no interference of vegetation within the study area. Because $P_F$ is very small compared with $P_N$ in regions of high moisture saturation, such as the CMRV [23], it was excluded from all analyses, and because of that we equals $P_T = P_N$. To estimate $P_N$, we installed 13 rain gauges under the canopy, provided with a 1 m long and 15.24 cm wide PVC channel to the trunk of each individual tree. The rain gauges were equipped with a tipping bucket rain gauge and a micro-data logger (Hobo Event, Onset Computer Corporation, Bourne, Miami, FL, USA), recording the time of each precipitation event. All rain gauges were calibrated by measuring the number of pulses per liter of water independently for each rain gauge (see more details in [32]).

The rain gauges were placed in the SW–NE direction, along the mountain slope based on the position of the individual trees and land availability. The geographic position of each individual was obtained with a global positioning system (GPS GARMIN eTrex 22 X, Garmin Corporation, Olathe, KS, USA). The external rain gauge was installed in a flat area in the northern part of the study area (Figure 1; in Supplementary Table S2, we present the location and characteristics of each individual tree).

### 2.3. Precipitation Types

The type of precipitation event was determined based on the amount of $P_G$ and $P_N$ as follows:

$$\text{Type 1: } P_G > P_N \qquad \text{Precipitation} \tag{2}$$

$$\text{Type 2: } P_G < P_N \qquad \text{Fog with precipitation} \tag{3}$$

$$\text{Type 3: } P_G < P_N, \text{ with } P_G = 0 \qquad \text{Fog} \tag{4}$$

A type 1 event (hereafter $T_1$) was characterized by a greater amount of $P_G$, and in many cases, this was the only type of precipitation present. In $T_1$, the occurrence of fog was uncertain; therefore, it was considered as a precipitation event only. In type 2 events ($T_2$), indirect rainfall was registered. This extra water supply was observed in a higher value of water under the canopy than outside of it. $T_2$ was characterized by the presence of fog and rain at the same time, where the fog was considered an extra source of water to the system. Finally, in type 3 events ($T_3$), $P_G$ was zero. This indicated the presence of fog. As fog moved horizontally by the effect of wind, it was deposited in the canopy and then precipitated on the ground and was captured by the rain gauges. In $T_3$, the external rain gauge did not detect fog due to its horizontal movement.

*2.4. Water Balance*

The water used by plants was determined by their transpiration rates. When the relative humidity was 100%, plants stopped transpiration and absorption of water from the soil, causing water saving. However, if evapotranspiration occurred, this represented a loss of water when this was greater than the transpiration [33,34]. To assess if water was stored in the canopy or lost during a precipitation event, it was necessary to know the potential transpiration of the plant (TRP) during the event, as well as the time when leaves remained wet after the event [35]. The water balance can be described as [36]

$$\theta + R = P_T + C - ET \tag{5}$$

where $\theta$ is the storage of groundwater in and below the surface, $P_T$ is the total precipitation, $C$ is the condensation as dew, or in our case fog precipitation, $ET$ is evapotranspiration, and $R$ is runoff—runoff was not measured for this work since our objectives only referred to the effect of vegetation on the input of water by precipitation and fog was taken as part of storage. Positive signs indicated the addition of water to the system, while negative signs indicated the consumption of water by vegetation. The total precipitation ($P_T$) was composed of the sum of $P_G$ and fog; nevertheless, both elements were intercepted by vegetation, which generated a loss of $P_G$ and a contribution of water originated by the fog. Thus, the contribution of $P_G$ remained as $P_G$ minus the interception and was equivalent to the net rain precipitation ($P_{NR}$). As for fog, the amount of water intercepted by the canopy and subsequently precipitated represents an addition to the system, which is referred to as the net fog precipitation ($P_{NF}$).

During a precipitation or fog event, water on the canopy evaporated, limiting transpiration; thus, TRP was considered as water saving by diminishing the absorption of water required for $ET$. However, in some cases, the water on the leaves is not sufficient to cover the $ET$ requirements, so plants need to absorb groundwater [33]. This amount of water is identified as $\eta$, and it refers to water that vegetation needs to absorb to cover ET requirements and represents a water loss. The equation to estimate the storage of underground water is

$$\theta = P_{NR} + P_{NF} + TRP - \eta + C \tag{6}$$

and $P_{NF}$ is estimated by the relation

$$P_{NF} = P_N - P_{NR} \tag{7}$$

where $P_N$ is the total net precipitation that was measured by the rain gauges and $P_{NR}$ is the net rain precipitation. In $T_1$, $P_{NR}$ was equivalent to $P_N$; in $T_3$, it was not present; and in $T_2$, it was not possible to know it directly since rain and fog occurred in the same event; here it was necessary to know the relation between $P_G$ and its interception. Based on the results of $T_1$, we obtained an equation that approximates the relation between $P_G$ and fog interception ($I_F$):

$$P_{NR} = -0.0207\, P_G{}^2 + 0.7871\, P_G - 1.4377 \tag{8}$$

### 2.5. Transpiration (TRP) and Evapotranspiration (ET)

Transpiration was estimated measuring the sap flow in the trunk using steady-state xylem water mass flow metering systems similar to those described by Čermáck et al. [37] and Schulze et al. [38] (Dynagage, SGB150-WS, Dynamax, Inc. Houston TX, USA) with one instrumental set per tree; additionally, it was required to determine the time period when the leaves remained wet. For this, we calculated the potential evapotranspiration (*ET*) reducing the Penman–Monteith model when stomatal resistance is null [39]:

$$ET = [\Delta Q_N + \rho_a C_p (VPD/r_A)]/[\lambda(\Delta + \gamma)] \tag{9}$$

where $\lambda$ is the latent heat of vaporization (kJ kg$^{-1}$), $\Delta$ is the slope of the saturation vapor pressure (kPa $°C^{-1}$), $Q_N$ is the net radiation flux (W m$^{-2}$), $\rho_a$ is the air density at constant pressure (kPa K$^{-1}$), Cp is the specific heat of air at constant pressure (J kg$^{-1}$ K$^{-1}$), VPD is vapor pressure deficit of the air (kPa), $\gamma$ is the psychrometric constant, and $r_A$ is the aerodynamic resistance (s m$^{-1}$). Vapor pressure deficit is calculated as VPD = $e_S$(1−RH), where $e_S$ is the saturation vapor pressure and RH is the relative humidity. Aerodynamic resistance ($r_A$) was estimated as

$$r_A = [1/(k^2)(u)] \ln [(Z_W - d)/Z_0] \ln [(Z_W - d)/(0.2)(Z_0)] \tag{10}$$

where $Z_W$ is the height at which the wind was measured (m); *d* is the level of the displacement of zero (i.e., the height of the active surface that determines the degree of deviation of the volume of air exerted by the vegetation [m]); *k* is the von Karmann constant (*k* = 0.41); *u* is the wind speed (m s$^{-1}$); and $Z_0$ is a measure of the aerodynamic heterogeneity of the surface, in this case, of the vegetation (m). *d* is calculated as

$$d = 0.245\,h + 0.091\,LAI \tag{11}$$

and $Z_0$ is calculated as

$$Z_0 = 0.0275 + 0.291\,h - 0.028\,LAI \tag{12}$$

where *h* is the height of the canopy (m) and LAI is the leaf area index (mm$^{-2}$), which for the study area was 4.7 m/m$^2$ [40].

For our analysis, the evapotranspiration intensity (mm/s) was not used as a parameter in the water balance; instead, we used the total volume of evapotranspiration and condensation (mm).

### 2.6. Microclimate Variables

The values of net radiation ($Q_N$), air temperature ($T_A$), humidity (RH), and wind speed (WS) and direction (WD) were obtained at the La Joya automatic station with a net radiometer (NR-Lite2, Campbell Scientific, Logan, UT, USA), a temperature–humidity probe (HMP35C, Campbell-Scientific), and an anemometer and vane set (03001, RM Young, Traverse City, MI, USA), respectively. Instruments were connected to a data logger (21X, Campbell Scientific, Logan, UT, USA) and scanned every second, and average values were logged every 20 min.

Once we obtained all the values of the given parameters, we estimated *ET*. A negative value of *ET* implied condensation of water on the leaf surface, which generated a gain of water since it limited the TRP; it also represented an additional amount of water to the system [24]. *ET* values were given only for events when $P_N$ occurred, regardless of the time that the leaf remained wet at the end of each event.

### 2.7. Water Gain and Loss

The amount of water absorbed by leaves was determined as

$$\delta = I_T - ET \tag{13}$$

where δ is the intercepted water that is stored in leaves at the end of a precipitation event. A positive δ implied that *ET* was not sufficient to evaporate all the water retained by the canopy. At the end of an event, some of the intercepted water remained on the vegetation cover increasing the time when the TRP was limited. Conversely, a negative δ generated by an *ET* value greater than the amount of water intercepted by the canopy implied the absorption of groundwater by the plant, representing a loss of water to the system. This loss was represented as η, to make a distinction with positive δ.

For fog events, it was not possible to obtain these values since it was necessary to know the moment when the precipitation event ended to determine δ. In addition, the installed rain gauges were not designed to capture horizontal precipitation, only $P_N$; thus, it was not possible to obtain the beginning and end references for fog events.

To know the time that δ took to evaporate ($t_E$), we used the *ET* value of the last hour of the event in which it was originated. Thus, the total time leaves remained wet ($t_H$) can be estimated with the following relation:

$$t_H = t_E + D_E \tag{14}$$

where $D_E$ is the duration of the event. If an event occurred before the stored water was completely evaporated, we used only the time elapsed between the end of the event that originated the water storage and the beginning of the next event. The remaining water added to the next event was not considered as part of the following event; however, the volume of water was calculated and identified as $\alpha_E$.

After estimating $t_H$, we calculated the amount of water that was not absorbed from the soil by the plant by estimating the amount of water transpired during the time when leaves remained wet. We considered this as the potential transpiration (TRP) since it was the water that vegetation would have transpired in the absence of a precipitation event.

*2.8. Fog*

The total volume of fog water was calculated with the equation

$$N = P_{NF} + ET_N + C_N \tag{15}$$

where N is the total volume of fog, $ET_N$ is the amount of fog intercepted and subsequently evaporated, and $C_N$ is the amount of water absorbed by the leaves after being intercepted. $ET_N$ was estimated as follows:

$$ET_N = ET - ET_R \tag{16}$$

where $ET_R$ is the amount of rainwater evaporated after being intercepted, considering that the proportions between $P_N$ and $P_G$ are equal to the ET ratios for each type of event:

$$ET_R = (P_{NN}/P_N) \, ET \tag{17}$$

It is also possible to know the total interception of precipitation, regardless of its origin, using the relation

$$I_T = I_F + I_R \tag{18}$$

where $I_N$ can be determined by the form

$$I_F = C_N + ET_N \tag{19}$$

*2.9. Statistical Analysis*

Statistical analyses were conducted using the software R [41]. We used the non-parametrical test Kruskal–Wallis to evaluate significant differences when we compared precipitation-type events: (1) the total number of events, (2) the duration in hours, (3) the gross precipitation, (4) the net precipitation, (5) interception, (6) transpiration, (7) evapotranspiration, and (8) the use of water by vegetation. We also compared the (1) net precipitation,

(2) gross precipitation, and (3) fog volume between seasons (wet vs. dry). The statistical significance was considered at 95% for all cases.

## 3. Results

### 3.1. Events

We registered 510 precipitation events (hereafter events) distributed throughout the year with three peaks during October (78 events), May (68 events), and January (52 events). Conversely, November and March had the lowest number of events with 24 and 23, respectively (Supplementary Table S3). We recorded 225 precipitation events during the dry season (1.2 events per day) and 285 during the wet season (1.6 events per day), with an average of 1.4 events per day throughout the study period (2006–2007). Regarding the types of events, $T_3$ events were 3.1 times greater than $T_1$ and $T_2$; $T_3$ had 386 events (76%), followed by $T_2$ with 83 events (16%) and $T_1$ with 41 events (8%) (Figure 2). The total duration was 2996 h, with 1600 h of $T_2$ (53%; with its maximum in November), 1342 h of $T_3$ (45%; maximum in January), and 54 h of $T_1$ (2%; maximum in August) (Figure 2; detailed data per month are given in Supplementary Table S4).

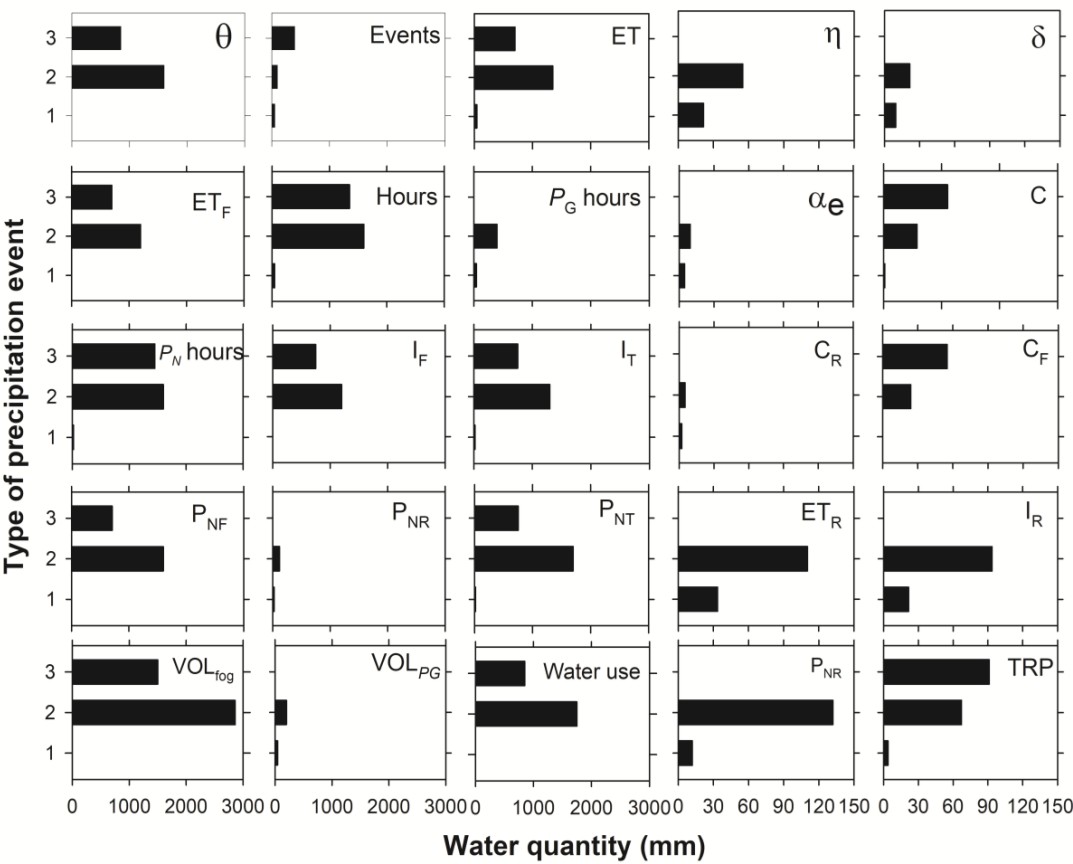

**Figure 2.** Values of water used ($\theta$; mm), total number of precipitation events (Events), total evapotranspiration (ET; mm), water absorbed ($\eta$; mm), water stored ($\delta$; mm), fog evapotranspiration (ET$_F$; mm), total number of hours of precipitation (Hours), total number of hours of gross precipitation (Hours $P_G$), water for extra storage ($\alpha_e$; mm), total condensation (C; mm), total number of hours of net precipitation (Hours $P_N$; mm), fog interception (I$_F$; mm), total interception (I$_T$; mm), rainfall condensation (C$_R$; mm), fog condensation (C$_F$; mm), net precipitation generated by fog ($P_{NF}$; mm), net precipitation generated by rainfall ($P_{NR}$; mm), total net precipitation ($P_{NT}$; mm), rain evapotranspiration (ET$_R$; mm), rain interception (I$_R$; mm), volume of fog (Vol$_{fog}$; mm), volume of gross precipitation (Vol$_{PG}$; mm), water consumed by the plant (Water use; mm), net precipitation generated by rainfall ($P_{NR}$; mm), and total transpiration (TRP; mm). Data are given according to the type of precipitation event (1, 2, and 3; see details in Methods).

We recorded 386 fog events, 41 rain events, and 83 events combining fog and rain. Between May and July 2007, we recorded the highest number of events. $T_3$ predominated during all months (except November) (Figure 2; Supplementary Table S3). The total precipitation recorded (i.e., $P_G$ + fog) was 4508.6 mm, with an average of 8.8 mm/event, regardless of its origin. Rain contributed 197.5 mm, and fog contributed 4311.1 mm. The average precipitation per event was 1.6 mm/event, with 0.6 mm/event of rain exclusively and 2.1 mm/event in combination with fog.

### 3.2. Gross Precipitation

Gross precipitation ($P_G$) contributed 197.4 mm. $T_2$ contributed 170.9 mm (87%), $T_1$ provided 26.5 mm (13%), and $T_3$ had no contribution. $P_G$ mainly occurred during the wet season (115.7 mm), with a decrease in the dry season (81.9 mm). The maximum $P_G$ recorded was in September (33.2 mm), and the minimum was in March and May (0.2 mm) (Supplementary Table S3). We registered 485 h of $P_G$; 50 h corresponded to $T_1$ (maximum in August), and the remaining 435 h corresponded to $T_2$ (maximum in October) (Supplementary Table S4).

### 3.3. Fog

We estimated a total of 4311.1 mm contributed by fog; 2820.8 mm corresponded to $T_2$ (maximum in September and minimum in March), and 1490.4 mm corresponded to $T_3$ (maximum in September and minimum in November). The $T_1$ events did not provide fog (Figure 2; Supplementary Table S3).

### 3.4. Net Precipitation

The precipitation generated in total by $P_G$ and fog was 2452.4 mm; 95.5% corresponded to fog, and 4.5% corresponded to $P_G$ (Supplementary Table S5). $P_N$ was higher during the wet season (1535.6 mm; maximum in September) (Supplementary Table S3). $T_3$ contributed with 739.5 mm (30.15%), $T_2$ contributed with 1703.3 mm (69.46%), and $T_1$ contributed with 9.6 mm (0.39%) (Figure 2). We recorded 2954 h; of these, 1589 h corresponded to $T_2$ (maximum in July), 1342 h corresponded to $T_3$ (maximum in January), and 23 h corresponded to $T_1$ (maximum in August) (Supplementary Table S4).

### 3.5. Interception

The total interception of precipitation ($I_T$) was 2053.5 mm; 1966.4 mm corresponded to intercepted fog ($I_F$), whereas 87.1 mm corresponded to intercepted rain ($I_R$). From the total $I_F$, $T_3$ contributed 1215.5 mm, and $T_2$ contributed 750.9 mm. Regarding $I_R$, 16.9 mm were captured from $T_1$, and 70.2 mm were captured from $T_2$ (Figure 2; Supplementary Table S5). We recorded the highest interception in September (338.7 mm) and the lowest in March (23.5 mm) (Supplementary Table S6). The months with the highest interception were August for $T_1$ (11.9 mm), September for $T_2$ (258.9 mm), and January for $T_3$ (134.1 mm) (Supplementary Table S6). Of the $I_T$, 96.9% was evapotranspirated, and 3.1% was condensed. $I_R$ represented 44% of water loss absorbed by the soil. Additionally, we found that 55.9% of $P_G$ reached the soil through the vegetal canopy, while 44.1% was lost by interception. Regarding $P_N$, 54.3% was precipitated after being intercepted by vegetation, whereas 45.7% was retained in the canopy preventing TRP; thus, more than half of the fog precipitation was incorporated into the soil after trees intercepted it. Overall, from the total precipitation ($P_T$), trees intercepted 45.5%, while 54.5% was subsequently precipitated as $P_N$.

During the dry season, the amount of fog intercepted was higher (1126.2 mm) than during the wet season (840.2 mm). In February, the intercepted fog was the highest (292.9 mm), whereas in March, we registered the lowest (23.2 mm) (Supplementary Table S6). Similarly, $I_R$ was lower in the wet season (51.2 mm), compared with the dry season (359 mm). September had the highest intercepted $P_G$ (13.8 mm) and the lowest was recorded in March

(0.2 mm). $T_2$ contributed the highest intercepted $I_R$ (70.2 mm) compared with $T_1$ (16.9 mm) (Supplementary Table S6).

### 3.6. Evapotranspiration

The average $ET$ was $1.9 \times 10^{-4}$ mm s$^{-1}$, and the total $ET$ was 2013.7 mm. Condensation ($C$) was $4.34 \times 10^{-5}$ mm s$^{-1}$, with a total volume of 2013.7 mm evapotranspirated and 64.3 mm condensed (Supplementary Table S7). From the total of evapotranspiration, 1279.7 mm corresponded to $T_2$, 709.3 mm corresponded to $T_3$, and 24.7 mm corresponded to $T_1$ (Figure 2; Supplementary Table S7). Regarding condensation, $T_3$ contributed 41.6 mm, $T_2$ contributed 21.3 mm, and $T_1$ contributed 1.4 mm (Figure 2; Supplementary Table S7).

The average $ET$ had its maximum in September ($2.8 \times 10^{-4}$ mm s$^{-1}$) and minimum in June ($6.4 \times 10^{-5}$ mm s$^{-1}$). In the wet season, we recorded an average $ET$ of $1.8 \times 10^{-4}$ mm s$^{-1}$, and during the dry season, the average was $1.8 \times 10^{-4}$ mm s$^{-1}$ (Supplementary Table S8). $T_1$ had maximum and minimum averages during October ($3.1 \times 10^{-4}$ mm s$^{-1}$) and June ($3.6 \times 10^{-5}$ mm s$^{-1}$), respectively. The maximum and minimum averages for $T_2$ occurred in September ($3.2 \times 10^{-4}$ mm s$^{-1}$) and June ($2.7 \times 10^{-5}$ mm s$^{-1}$), respectively, and for $T_3$, the maximum and minimum averages occurred in September ($3.4 \times 10^{-4}$ mm s$^{-1}$) and March ($5.9 \times 10^{-5}$ mm s$^{-1}$), respectively (Supplementary Table S8). As for the $ET$ volume, the maximum amount was registered during the wet season (862.4 mm) with the maximum in September (325.7 mm). During the dry season, we found the highest volume of $ET$ (1151.3 mm) (Supplementary Table S8).

During the wet season, $ET$ from fog was 818.2 mm, while during the dry season, it was 1088 mm, with the maximum in September (316.8 mm) and minimum in March (22.6 mm) (Supplementary Table S9). The fog evaporated by $T_2$ was 471.3 mm (maximum in September, 241.4 mm; minimum in March, 0.9 mm), whereas for $T_3$, the maximum was in January (127.8 mm), and the minimum was in March (21.7 mm) (Supplementary Table S9).

### 3.7. Stored Water

The total amount of stored water was 25.3 mm, mostly contributed by $T_2$ (17.4 mm) compared with $T_1$ (7.9 mm); 19 mm were stored in the wet season, and 6.3 mm were stored in the dry season (Table 1). The stored water needed for $ET$ was 57.9 mm, of which 16.3 mm was stored during $T_1$ and 41.7 mm was stored during $T_2$ (Table 1). The time required for the stored water to evaporate was 67.5 h; 53.4 h was required to evaporate the water contributed by $T_2$, and 14.1 h was required for $T_1$. A total of 47.6 h in the wet season and 19.9 h in the dry season was required to evaporate the stored water.

**Table 1.** Total amount of water stored to be evaporated ($\delta$, mm) and water absorbed by plants ($\eta$, mm) during the wet (Wet) and dry (Dry) seasons. Data are given in mm according to the type of precipitation event ($T_1$ and $T_2$; see details in Materials and Methods).

| | $\delta$ | | $\eta$ | |
|---|---|---|---|---|
| **Type** | **Wet** | **Dry** | **Wet** | **Dry** |
| $T_1$ | 5.97 | 1.99 | 9.50 | 6.77 |
| $T_2$ | 13.07 | 4.29 | 12.52 | 29.17 |
| Total | 19.04 | 6.28 | 22.02 | 35.93 |

### 3.8. Transpiration

The total amount of transpired water was 118.5 mm, with the maximum in January (14.4 mm) (Figure 2). During the dry season, the TRP was limited to 62.5 mm, while during the wet season, it was 56 mm (Supplementary Table S9). $T_1$ prevented the TRP of 2.7 mm, whereas $T_2$ prevented the TRP of 47.5 mm, being similar for both seasons (wet: 23.5 mm vs. dry: 23.9 mm). During the wet season, $T_3$ prevented the TRP of 37.4 mm and 30.9 mm during the dry season (Supplementary Table S9).

*3.9. Water Balance*

A total of 2577.3 mm of water was not absorbed by the soil, contributed by $T_2$ and $T_3$ (1730.4 mm and 849.5 mm, respectively). $T_1$ represented a water loss to the system with 2.6 mm consumed. September had the highest amount of water saved in the soil (752.7 mm), while March and May had the minimum (31.7 mm and 31.9 mm, respectively). During the wet season, we registered the highest amount of water saved (1600.2 mm), contrasting to the dry season (977.1 mm) (Figure 3; Supplementary Table S9).

Regarding monthly water consumption, the highest loss for $T_1$ was in October ($-4.8$ mm), while in August, we observed the highest amount of saved water (9.9 mm). No water loss occurred during the wet season, while $-4.4$ mm were lost during the dry season. For $T_2$, the highest amount of saved water occurred during the wet season (1022.2 mm; maximum in September), in contrast to the dry season (708.1 mm). $T_3$ had its maximum in September (240.1 mm) and minimum in November (14.8 mm). In the wet season, the saved water was 576 mm, and in the dry season, the saved water was 273.5 mm (Figure 3; Supplementary Table S9).

As for condensation (C), this was higher for $T_3$ (41.6 mm), followed by $T_2$ (21.3 mm) and $T_1$ (1.4 mm). The condensation caused by fog ($C_F$) was also the highest for $T_3$, but the highest condensation caused by rain ($C_R$) corresponded to $T_2$. In addition, $T_2$ had the highest values of water for storage ($\delta$), absorption ($\eta$), consumption ($\theta$), and extra storage (Figure 2; Table 2).

**Table 2.** Values of total (C), average ($C_A$), fog ($C_F$), and rainfall ($C_R$) condensation; transpiration (TRP); and water stored ($\delta$), absorbed ($\eta$), used ($\theta$), and for extra storage ($\alpha_e$). Data are given in mm according to the type of precipitation event ($T_1$, $T_2$, and $T_3$; see details in Materials and Methods).

| Type | C | $C_A$ | $C_F$ | $C_R$ | TRP | $\delta$ | $\eta$ | $\theta$ | $\alpha_e$ |
|------|------|---------------------|-------|-------|--------|-------|-------|---------|-------|
| $T_1$ | 1.42 | $5.6 \times 10^{-05}$ | 0 | 1.42 | 2.68 | 7.95 | 16.27 | $-29.05$ | 4.48 |
| $T_2$ | 21.26 | $4.0 \times 10^{-05}$ | 18.59 | 2.67 | 47.45 | 17.36 | 41.68 | 1559.4 | 6.82 |
| $T_3$ | 41.64 | $4.3 \times 10^{-05}$ | 41.64 | 0 | 68.38 | 0 | 0 | 849.49 | 0 |
| Total | 64.33 | $4.3 \times 10^{-05}$ | 60.24 | 4.09 | 118.51 | 25.32 | 57.95 | 2379.8 | 11.3 |

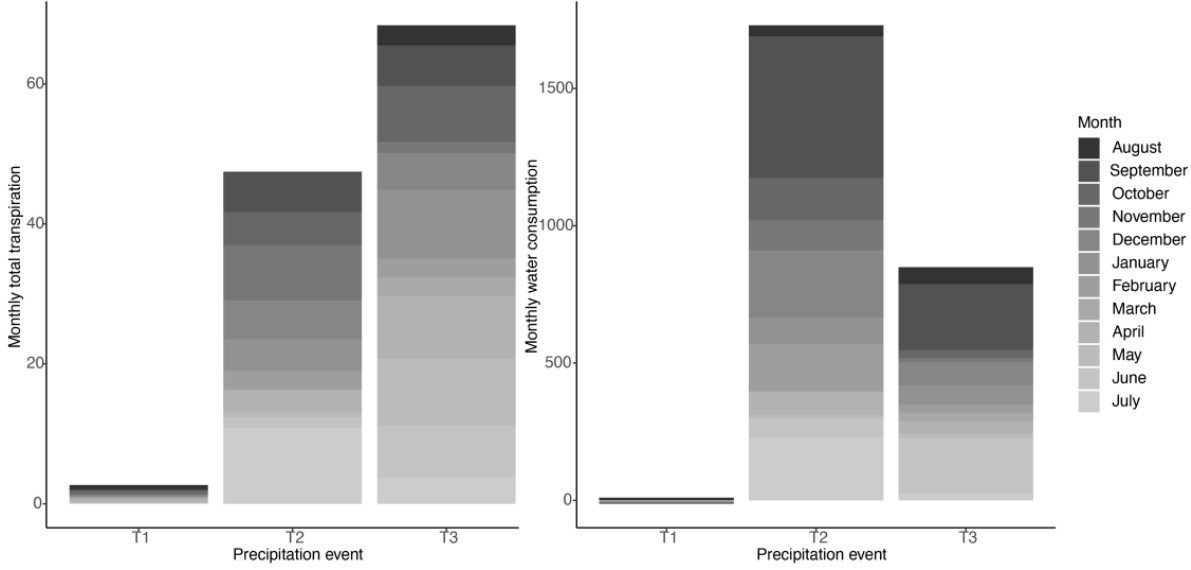

**Figure 3.** Monthly values of total transpiration (TRP) and water consumption ($W_C$). Data are given in mm according to the type of precipitation event ($T_1$, $T_2$, and $T_3$; see details in Materials and Methods).

### 3.10. Statistical Differences

We found significant differences when we compared the types of events for the total number of events, duration in hours, gross precipitation, net precipitation, interception, transpiration, evapotranspiration, and use of water. We found no significant differences when we compared the seasons for net precipitation, gross precipitation, and fog. Finally, we found significant differences when we compared the volumes of gross precipitation and fog (Table 3).

**Table 3.** Kruskal–Wallis results evaluating significant differences among precipitation-type events for total number of events, duration in hours, gross precipitation, net precipitation, interception, transpiration, evapotranspiration, and use of water by vegetation; between seasons (wet vs. dry) for net precipitation and gross precipitation; and among the volume of gross precipitation and fog volume.

| Comparison | Variable | *H* Value | *p* Value |
|---|---|---|---|
| Type of event | Total number of events | 24.111 | *p* < 0.001 |
| | Duration in hours | 21.206 | *p* < 0.001 |
| | Gross precipitation | 4.7183 | *p* = 0.029 |
| | Net precipitation | 24.259 | *p* < 0.001 |
| | Interception | 20.8 | *p* < 0.001 |
| | Transpiration | 21.105 | *p* < 0.001 |
| | Evapotranspiration | 17.251 | *p* < 0.001 |
| | Use of water | 24.083 | *p* < 0.001 |
| Seasons | Net precipitation | 0.641 | *p* = 0.4 |
| | Gross precipitation | 2.5731 | *p* = 0.1 |
| | Fog | 0.10256 | *p* = 0.7 |
| Volume of gross precipitation | Fog | 17.288 | *p* < 0.001 |

## 4. Discussion

Our results highlight the importance of fog as a water supply (fog increased 24 times the amount of water available during this season) in the Central Mountain Region of Veracruz of Eastern Mexico. The importance of fog has been broadly acknowledged in previous studies (e.g., [42–45]). Fog is recognized as an important contributor of water to ecosystems, although its assessment and quantification are not always easy to determine and have limitations [46].

We also found the importance of vegetation to increase the stored water in the cloud forest. In this ecosystem, the presence of fog and the high relative humidity of the air are characteristic parameters [15]. These two parameters, generally related to low temperatures, keep cloud forests permanently humid, favoring the presence of epiphytes (e.g., mosses and lichens) that can keep microclimatic humidity, even though when, at the macroclimatic level, the relative humidity decreases [47,48]. Thus, the presence of vegetation significantly alters the atmospheric properties near the soil [24].

In the study area, fog had a substantial contribution of humidity, with a total volume almost 22 times greater than that of rainfall (4311.1 mm was contributed by the fog, and 197.5 mm was contributed by rainfall). The fog presence had a direct water contribution to the system. Almost half of the total volume of water corresponded to fog (45.6%). This water was intercepted and precipitated later to the ground, representing an input for groundwater. Fog represented a 33% savings in groundwater consumption, and 44.4% of the fog volume was absorbed by the soil to compensate the TRP; in some cases, it generated an extra water supply. Fog also increased the time foliage remained wet by 2.3%.

Fog can be considered a constant source of moisture throughout the year in the study area. However, water intercepted during the dry season was higher (56%, and mainly during February) compared with the wet season (36%). Several studies have shown the importance of fog during the dry season as a source of water (e.g., [42,45,49,50]) and, importantly, as a supply of drinking water for local communities (e.g., [15,51]). Further,

Fischer et al. [45] showed the importance of fog on different aspects of tree functions and the implications for future climate change. This aspect is relevant for our region since increases in temperature and reductions in precipitation are predicted to occur in the coming decades [28,29].

In the dry season, most of the fog was used for TRP, whereas in the wet season, most of it was directly absorbed by the soil when the vegetation intercepted it. Nevertheless, we acknowledge that these values may be higher for the region compared with other places since forests are generally capable of intercepting more water than other vegetation types [7]. The amount of water entering and passing through the canopy is determined by climatic factors, such as the water content of the clouds, the wind speed, and turbulence [52–54], and factors inherent to vegetation, such as the height and canopy structure—which influences the roughness, causes turbulence, and includes size, quantity, placement, and grouping of the foliage—and life forms and species of epiphytes [3,6]. Additionally, the spatial and temporal variation of interception in tropical and cloud forests can be very pronounced [55,56]. In particular, mountain and cloud forests have higher interception than other ecosystems (e.g., low humid tropical forests [17]). This is because the storage capacity varies according to the vegetation structure and canopy, which directly affects the amount of water intercepted [7,57]. By keeping the canopy wet, the amount of water absorbed was reduced by half, implying significant water savings to the system.

Our findings provide evidence of the importance of vegetation in the water balance. Alarmingly, Veracruz has been extensively deforested since the nineteenth century, and by the year 2000, only 18% of the total area maintained its original vegetation, and 8% had not been disturbed [58]. Deforestation implies an increase in flow rates due to the reduction of high $ET$ [59] and a reduction in groundwater recharge [60]. Rain and cloud forests, through their "sponge" effect, are of great hydrological importance, and their loss may have severe consequences for downstream valleys [61,62]. In the case of cloud forests in tropical regions, such as our study area, deforestation can cause a substantial loss of water in the basin. This is mainly due to the reduction of the additional income of water to the forests by horizontal precipitation, which affects the water balance of the system [59].

Fortunately, for our study area, we found a total gain of groundwater consumption of 2592.9 mm compared with the total loss of 15.7 mm (Table 2). Of the total water gain, the largest part (91%) was contributed by $I_F$, 4% was contributed by $P_{NR}$, 3% was contributed by the condensation of the water on leaves, and 2% was contributed by the water saved by limiting TRP. $I_R$ generated water loss by limiting the amount of water that reached the ground when it was intercepted by vegetation canopy; however, this volume was 165 times lower than the gain generated by $I_F$. Most of the water loss was caused by $I_R$, while the other 40% of the water was involved in $ET$. Importantly, the water involved in $ET$ has significant effects during periods of water deficit. When the foliage gets wet, the TRP decreases, and possible absorption of water through the foliar tissue can occur, preventing plants from suffering water stress [43]. When TRP stops or is limited, the absorption of water by the roots stops as well as increasing soil water [19].

Regarding the types of events, we found that $T_1$ generated a loss 1.2 times greater than the gain, whereas $T_2$ and $T_3$ represented a contribution of water. This implies that the vegetation presence might have generated a water loss to the system during the $P_G$ events; however, fog generated a gain. Our results show that $I_F$ had a contribution of 54% of water to the system directly through $P_{NF}$; for every water millimeter of fog, it contributed an average of 0.5 mm to the system.

Interestingly, we estimated that without forest vegetation, there would be a loss of 87.1 mm of water due to lack of rainfall interception ($I_R$); however, this value is not comparable to the gain that occurs by the fog in the area. Without fog, there would be a water loss of 2342.1 mm directly contributed to the ground, 1906.2 mm of $I_F$ limiting TRP, and 60.2 mm of $C$. Additionally, 2406.4 mm contributed directly into the system by $P_N$ and $C$, and 60.6 mm saved by limiting TRP would be lost. The presence of forest and fog in the region implied a decrease in the rate of groundwater consumption by vegetation

with a savings of 50% with respect to the amount of water that would be lost through TRP. Therefore, the presence of the forest–fog system is translated into a greater amount of water available. Moreover, if any of these two elements did not exist, 53% of the humidity that directly reaches the soil contributed by fog would be lost, and TRP would be considerably higher by reducing the time of wet foliage.

The conservation of the cloud forest, therefore, represents an important water and humidity supply to the region mainly caused by fog. The water contributed by fog can be potentially used for consumption, either human or animal, or industrial use. In a previous study near our study area, Barradas [23] found that the water collected by one individual of *Pinus montezumae* during a four-hour precipitation event could provide enough water to supply a family of four persons for more than a month.

It is important to know in more detail the process of water balance in the region, and we acknowledge some caveats emerging from our findings. We recognize the limitations of the size of our study area and the number of individual trees used; hence, we encourage studies using a greater area with different topographic characteristics and taking into consideration additional variables, such as the water runoff by the trunk, as well as developing analysis on the amount of water precipitated and absorbed by the soil. We also recommend conducting studies comparing species to assess which ones are more efficient at capturing precipitation. These findings could be used in afforestation programs aiming to increase local and regional precipitation. We also note that taking measurements related to water consumption in a pasture area would be beneficial in future studies since the region presents a large amount of grazing land and increasing deforestation [58]. We also suggest that future studies consider including fog capture measurements artificially, as well as an assessment of their possible use or storage. This assessment can help to have a better and more complete understanding of the water balance in the region and the possible implications of climate change.

## 5. Conclusions

We registered 510 precipitation events and 386 fog events throughout the year. During this time, fog contributed 4311.1 mm to the system, almost 22 times greater than the contribution of rainfall (197.5 mm), highlighting the importance of fog to the water supply in the region. Fog has a substantial water contribution to the system in the region of half of its volume, through the intercepted fog and, subsequently, water precipitated to the ground. While the other half is related to the limitation of water absorption of the soil to compensate for transpiration, limiting it and, in some cases, condensing and generating an extra supply of water, in addition to generating an increase in the time that the foliage remains wet, in relation to what would occur only during rain events. Likewise, fog represents an increase in the percentage of the amount of water available during the dry season, with respect to what could occur only with rainfall. Furthermore, fog is a constant source of humidity in the region since there is no considerable difference in its volume between dry and wet seasons.

In this way, the relationship between fog and vegetation in the region implies a decrease in the consumption of groundwater by fog limiting groundwater absorption and compensating plant transpiration, which implies a loss of water, becoming 50% of water savings. Therefore, although vegetation generates a decrease in the amount of rainwater that reaches the ground, fog can compensate for this loss by the water intercepted in the canopy, which is later precipitated to the ground, increasing water availability. The presence of the fog translates into a greater amount of water available in the region, and when fog or vegetation are not present, 53% of the humidity can be lost, and transpiration levels can increase because the period of leaf wetness is reduced.

**Supplementary Materials:** The following supporting information can be downloaded at: https://www.mdpi.com/article/10.3390/w15071286/s1, Table S1: Height, diameter, climatic requirements (temperature and precipitation ranges), and distribution (altitudinal range) of *Alnus acuminata*, *Cupressus benthamii*, *Crataegus mexicana*, *Pinus ayacahuite*, and *Pinus patula*; Table S2: Location and details of the individuals used for the rain gauge placement. See location in Figure 1; Table S3:

Monthly number of total precipitation events (Events) and volume of net precipitation ($P_N$), gross precipitation ($P_G$), and fog (Fog). Data are given in mm (except for events) according to the type of precipitation event ($T_1$, $T_2$, and $T_3$; see details in Materials and Methods); Table S4; Monthly total number of hours of total ($P$), gross ($P_G$), and net ($P_N$) precipitation. Data are given according to the type of precipitation event ($T_1$, $T_2$, and $T_3$; see details in Materials and Methods); Table S5: Values of total net ($P_N$), fog ($P_F$), and gross ($P_G$) precipitation and total ($I_T$), fog ($I_F$), and rain ($I_R$) interception. Data are given in mm according to the type of precipitation event ($T_1$, $T_2$, and $T_3$; see details in Materials and Methods); Table S6. Monthly values of total ($I_T$), fog ($I_F$), and rain ($I_R$) interception. Data are given in mm according to the type of precipitation event ($T_1$, $T_2$, and $T_3$; see details in Materials and Methods); Table S7. Values of average ($ET_A$), total ($ET$), fog ($ET_F$), and rain ($ET_R$) evapotranspiration and total ($C_T$), fog ($C_F$), and rain ($C_R$) condensation. Data are given according to the type of precipitation event ($T_1$, $T_2$, and $T_3$; see details in Materials and Methods); Table S8. Monthly average ($ET_A$) and volume ($ET_{vol}$) of evapotranspiration. Data are given in mm according to the type of precipitation event ($T_1$, $T_2$, and $T_3$; see details in Materials and Methods); Table S9. Monthly values of fog ($ET_F$) and rain ($ET_R$) evapotranspiration, total transpiration (TRP), and water consumption ($W_C$). Data are given in mm according to the type of precipitation event ($T_1$, $T_2$, and $T_3$; see details in Materials and Methods).

**Author Contributions:** Conceptualization, A.S.-F. and V.L.B., and J.C.-P.; methodology, A.S.-F. and J.C.-P. and V.L.B. and M.E.-R. and M.B.; software, M.E.-R.; formal analysis, M.E.-R. and M.B. and A.S.-F.; investigation, A.S.-F. and V.L.B.; resources, V.L.B. and J.C.-P.; writing—original draft preparation, M.E.-R. and V.L.B. and A.S.-F.; writing—review and editing, M.E.-R. and V.L.B.; supervision, V.L.B.; project administration, V.L.B.; funding acquisition, V.L.B. All authors have read and agreed to the published version of the manuscript.

**Funding:** This research was funded by Consejo Nacional de Ciencia y Tecnología y la Secretaría de Medio Ambiente y Recursos Naturales (CONACYT-SEMARNAT), Government of Mexico, grant number 2004-C01-0332, and the APC was funded by Instituto de Ecología, Universidad Nacional Autónoma de México through the operating fund.

**Data Availability Statement:** Not applicable.

**Conflicts of Interest:** The authors declare no conflict of interest.

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
