# Peer review of "How Important Are Fog and the Cloud Forest as a Water Supply in Eastern Mexico?"

_water, doi:10.3390/w15071286_

Round 1

Reviewer 1 Report

The paper is well written and reaches a number of conclusions that are relevant for water resources managers, especially at local and regional scales. All this being said, I must recommend minor revision for this paper.

Major comments:

I have only two major comments. Why authors used the national (in Spanish language) paper or report as a reference for introduction? For example, Line 39 (reference #3) for net precipitation, Line 43 (reference #4) for interception. Also, reference of paper is out of date, authors needs to update them. 

Minor comments:

L 43: What is the differences between evaporation and interception during precipitation?

L 45: Vegetation can decrease or increase the duration and quantity of rainfall (by throughfall or interception process, see for example: https://doi.org/10.1016/j.jhydrol.2005.03.021.

L 88: Add a climate of study area based on Koppen-Geiger classification.

L 101: net precipitation is throughfall+fog+stemflow, right? I mean, stemflow is the part of net precipitation (see for example, https://doi.org/10.1007/978-3-030-29702-2_4).

Author Response

Reviewer 1

Dear Reviewer 1

Thank you for your insightful comments and feedback to help us improve the quality of our work and strengthen its message. We have updated the manuscript following all comments and suggestions. Below we respond in detail to all your comments.

Yours sincerely,

The authors

 The paper is well written and reaches a number of conclusions that are relevant for water resources managers, especially at local and regional scales. All this being said, I must recommend minor revision for this paper.

            Response. We thank Reviewer 1 for their comments on the relevance of our study. We appreciate their comments and feedback.

 Major comments:

I have only two major comments. Why authors used the national (in Spanish language) paper or report as a reference for introduction? For example, Line 39 (reference #3) for net precipitation, Line 43 (reference #4) for interception. Also, reference of paper is out of date, authors needs to update them. 

            Response. We have updated the reference as suggested by Reviewer 1 and added the following references in the updated manuscript:

  • Badu M, Ghimire CP, Bruijnzeel LA, Nuberg I, Meyer WS. Net precipitation, infiltration and overland flow production in three types of community-managed forest in the Mid-hills of East Central Nepal. Trees, Forests and People. 2022. 8:100218.
  • Baiamonte G. Simplified Interception/Evaporation Model. Hydrology. 2021; 8(3):99. https://doi.org/10.3390/hydrology8030099
  • Fan J, Oestergaard KT, Guyot A, Lockington DA. Measuring and modeling rainfall interception losses by a native Banksia woodland and an exotic pine plantation in subtropical coastal Australia. Journal of hydrology. 2014. 515:156-65.
  • Keim RF, Skaugset AE, Weiler M. Temporal persistence of spatial patterns in throughfall. Journal of Hydrology. 2005. 314(1-4):263-74.
  • Robinson M, Ward RC. 2017. Hydrology: principles and processes. Iwa Publishing
  • Savenije HH. The importance of interception and why we should delete the term evapotranspiration from our vocabulary. Hydrological processes. 2004.18(8):1507-11.

Minor comments:

L 43: What is the differences between evaporation and interception during precipitation?

            Response: We thank Reviewer 1 for pointing out this lack of definitions in the original manuscript. We have added these definitions in the manuscript to avoid confusion. The updated manuscript reads as:

            For example, in a forest with a high leaf area index, a considerable part of the precipitation does not reach the ground because of interception (i.e., the part of rainfall that is inter-cepted by vegetation [7,8] or evaporation (i.e., the process of surface water absorbing heat energy from the sun or from warm air and changing from liquid to gas) [3,6].

L 45: Vegetation can decrease or increase the duration and quantity of rainfall (by throughfall or interception process, see for example: https://doi.org/10.1016/j.jhydrol.2005.03.021.

            Response: In light of this comment, we have updated the manuscript to include this point that was overlooked in the original manuscript. The updated manuscript reads as:

As for PG, this is the precipitation that reaches the top of the vegetation; however, the vegetation surface may have no effect or can decrease or increase the occurrence, quantity or duration of a precipitation event [9,10].

L 88: Add a climate of study area based on Koppen-Geiger classification.

            Response. We have added the climate classification as requested.

L 101: net precipitation is throughfall+fog+stemflow, right? I mean, stemflow is the part of net precipitation (see for example, https://doi.org/10.1007/978-3-030-29702-2_4).

Totally agree, however, as PF was excluded, throughfall was equal PN. Please see lines 107, 108 and 114 in the new manuscript and equation 1.

Reviewer 2 Report

General Comments

The authors designed the article to study the “How Important are Fog and the Cloud Forest as a Water Supply in Eastern Mexico?”. In this paper, authors are tried o answer the questions, i.e., 1) what is the fog-water contribution to the water balance? and 2) how does the presence of vegetation affect the water supply to the ecosystem?

The authors tried very well to explain their designed research. The following mentioned discrepancies are mentioned to increase the quality of the article before final publication.

Specific Comments

In line 50, the FG needs to be corrected.

Kindly clearly mention the novelty statement in the last paragraph of the introduction section.

Please mention all the instruments which are used to measure the different kind of water measurement instruments. i.e., transpiration, evapotranspiration, protentional evaporation, and fog etc.

My specific comment is about your result section because the most of the data is mentioned in the supplementary tables but very less results are mentioned in the introduction section. Try to convert the tabular data into the graphical form for better representation of the article results.

In table 1, kindly mention the units of the mentioned values.

In the 3.9 water balance heading, all the results are referred towards the supplementary data tables which are not mentioned in the main article. Therefore, try to the key results in the main article.

In the 3.10 heading, kindly try to mention the statistical differences in a better way and explain them in very detailed way.

The conclusion section needs to be revised with the quantification strategy.

Author Response

Reviewer 2

Dear Reviewer 2

Thank you for your insightful comments and feedback to help us improve the quality of our work and strengthen its message. We have updated the manuscript following all comments and suggestions. Below we respond in detail to all your comments.

Yours sincerely,

The authors

General Comments

The authors designed the article to study the “How Important are Fog and the Cloud Forest as a Water Supply in Eastern Mexico?”. In this paper, authors are tried o answer the questions, i.e., 1) what is the fog-water contribution to the water balance? and 2) how does the presence of vegetation affect the water supply to the ecosystem?

The authors tried very well to explain their designed research. The following mentioned discrepancies are mentioned to increase the quality of the article before final publication.

             Response. We appreciate Reviewer 2 comments and feedback. Following their suggestions, we have improved our manuscript.

Specific Comments

In line 50, the Fneeds to be corrected.

            Response. Thank you for noticing this mistake. It has been corrected in the updated manuscript.

Kindly clearly mention the novelty statement in the last paragraph of the introduction section.

            Response. We have added a novelty statement in the last paragraph of the introduction. The updated manuscript now reads as:

In this region, climate change and deforestation have led to a decrease in precipitation [27-29] representing a loss in the water balance and in the water supply to surrounding urban areas [19]. Despite of these changes in the region, little is known about the im-portance of fog in the water balance and its relationship with local vegetation. Our ob-jective was to answer two questions: (1) what is the water contribution of the fog to the water balance? And (2) how does the presence of vegetation affect this water supply?

Please mention all the instruments which are used to measure the different kind of water measurement instruments. i.e., transpiration, evapotranspiration, protential evaporation, and fog etc.

            Response. Instruments to measure PG, PN and fog, 1 rain gauge out the forest and 13 rain gauges under the forest canopy. Please see lines 111-118.

Transpiration was estimated measuring with Sap Flow sensors (Dynagage, SGB150-WS, Dynamax, Inc., Houston TX, USA), please see line 181.

Potential evapotranspiration: Please see lines 204-210

My specific comment is about your result section because the most of the data is mentioned in the supplementary tables but very less results are mentioned in the introduction section. Try to convert the tabular data into the graphical form for better representation of the article results.

            Response. We thank Reviewer 2 for this comment. Following their suggestions, we have updated the results section to improve our manuscript.

In table 1, kindly mention the units of the mentioned values.

            Response. We have added units (mm) in Table 1. Please se lines 365-366

In the 3.9 water balance heading, all the results are referred towards the supplementary data tables which are not mentioned in the main article. Therefore, try to the key results in the main article.

            Response. We have added a new figure in the updated manuscript to present results from Supplemental Table 9. See new Figure 3 in the updated manuscript.

In the 3.10 heading, kindly try to mention the statistical differences in a better way and explain them in very detailed way.

            Response. We have a table in the main text to describe the statistical differences to facilitate the reading. See new Table 3 in the updated manuscript.

The conclusion section needs to be revised with the quantification strategy.

            Response. Following this suggestion, we have added quantitative results in the conclusion. The updated manuscript now reads as:

We registered 510 precipitation events and 386 fog events throughout the year. During this time, fog contributed 4311.1 mm to the system; almost 22 times greater than that of rainfall (197.5 mm), highlighting the importance of fog to the water supply in the region. Fog has a substantial water contribution to the system in the region of half of its volume, through the intercepted fog and subsequently, water precipitated to the ground.
